# MedSyn: Text-guided Anatomy-aware Synthesis of High-Fidelity 3D CT Images

**Yanwu Xu[1*], Li Sun[1*], Wei Peng[2*], Shyam Visweswaran[3], and Kayhan Batmanghelich[1]**

[1] Boston University, [2] Stanford University, [3] University of Pittsburgh
{yanwuxu, lisun, batman}@bu.edu, wepeng@stanford.edu, shv3@pitt.edu

## Abstract

This paper introduces an innovative methodology for producing high-quality 3D lung CT images guided by textual information. While diffusion-based generative models are increasingly used in medical imaging, current state-of-the-art approaches are limited to low-resolution outputs and underutilize radiology reports' abundant information. The radiology reports can enhance the generation process by providing additional guidance and offering fine-grained control over the synthesis of images. Nevertheless, expanding text-guided generation to high-resolution 3D images poses significant memory and anatomical detail-preserving challenges. Addressing the memory issue, we introduce a hierarchical scheme that uses a modified UNet architecture. We start by synthesizing low-resolution images conditioned on the text, serving as a foundation for subsequent generators for complete volumetric data. To ensure the anatomical plausibility of the generated samples, we provide further guidance by generating vascular, airway, and lobular segmentation masks in conjunction with the CT images. The model demonstrates the capability to use textual input and segmentation tasks to generate synthesized images. The results of comparative assessments indicate that our approach exhibits superior performance compared to the most advanced models based on GAN and diffusion techniques, especially in accurately retaining crucial anatomical features such as fissure lines, airways, and vascular structures. This innovation introduces novel possibilities. This study focuses on two main objectives: (1) the development of a method for creating images based on textual prompts and anatomical components, and (2) the capability to generate new images conditioning on anatomical elements. The advancements in image generation can be applied to enhance numerous downstream tasks.

## Introduction

Denoising Diffusion Probabilistic Models (DDPM) (Ho, Jain, and Abbeel 2020) and score-based generative models have become prominent in computer vision and medical imaging, noted for their stable training and high-quality outputs. Advanced tools like IMAGEN (Saharia et al. 2022) and latent Diffusion Models (LDMs) (Rombach et al. 2022) now enable fine-grained image generation guided by text

---

prompts, which is beneficial for medical imaging tasks such as data generation and uncertainty quantification. Although methods such as RoentGen (Chambon et al. 2022) have illustrated the potential of 2D cross-modality generative models conditioned on text prompts. The field lacks text-guided 3D volumetric medical image generation techniques, a challenge due to memory constraints and the need to preserve anatomical details. This paper seeks to address these challenges.

Enhancing resolution in generative models, particularly for 3D images, demands significant memory, as neural networks require large capacities to store information and maintain anatomical accuracy while responding to text prompts. Existing methods like GANs (Goodfellow et al. 2014) and conditional Denoising Diffusion Probabilistic Models (cDPMs) (Peng et al. 2022) face their own limitations, such as reduced sample diversity or high memory usage. Moreover, integrating text prompts, like radiology reports, into image generation requires high-capacity networks to depict complex medical pathologies accurately.

In 3D medical imaging, inaccuracies or "hallucinations" can result in biased or incorrect representations, a critical concern in medical contexts. We focus on CT lung imaging, where such inaccuracies can lead to serious misrepresentations. Unlike 2D X-rays, 3D imaging exacerbates these issues, making training on extensive datasets impractical and necessitating strong model constraints.

We introduce MedSyn, shown in Figure 1, a novel approach for high-resolution, text-guided, anatomy-aware volumetric image generation to overcome these obstacles. Utilizing around 9,000 3D chest CT scans with accompanying radiology reports, our method starts with low-resolution synthesis and progresses to higher resolutions. We've enhanced the UNet's capacity without significantly increasing memory demands, allowing for textual guidance in synthesis. Our method also includes segmentation masks for detailed anatomical structures, ensuring accuracy in the generated images. Comparative experiments demonstrate the superiority of our approach in terms of generative quality and efficiency, emphasizing the importance of our method's innovative components.

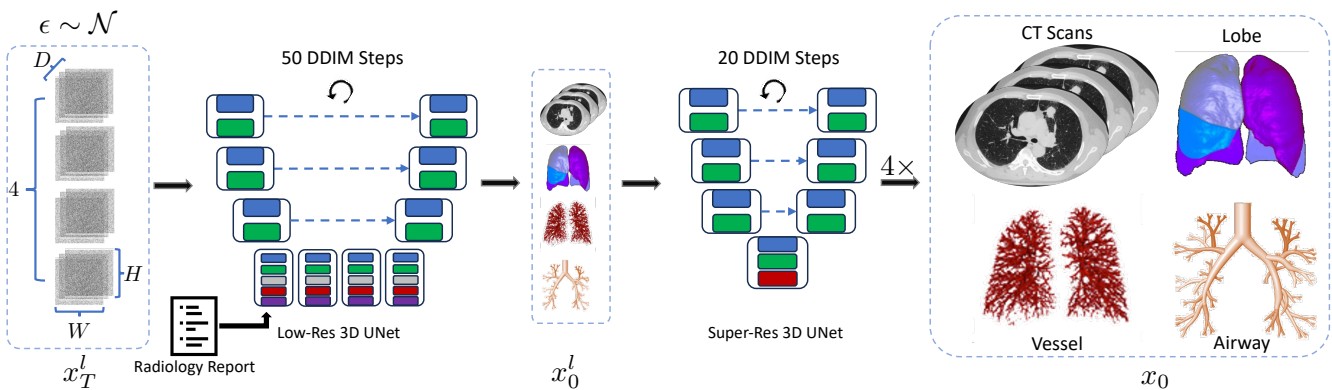

Figure 1: Overview of our generative model, MedSyn. Using a hierarchical approach, we first generate a $64 \times 64 \times 64$ low-resolution volume, along with its anatomical components, conditioning on Gaussian noise $\epsilon$ and radiology report. The low-resolution volumes are then seamlessly upscaled to a detailed $256 \times 256 \times 256$ resolution.

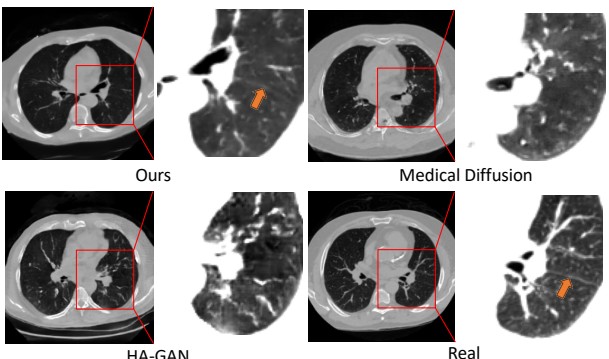

Figure 2: Randomly generated images (from HA-GAN and Medical Diffusion) and the real images. Our method is the only one that can preserve delicate anatomical details, including fissures, as indicated by the arrows.

## Method

Our radiology report conditional generated framework (MedSyn), as illustrated in Fig. 1, is a hierarchical model built upon the conditional diffusion models, which take the inputs of a random Gaussian noise $\epsilon$ and the text token features $f_{text}$ to generate a $256 \times 256 \times 256$ medical volumes. The proposed model has three core components: 1. A pre-trained text-encoder (Medical BERT) (Alsentzer et al. 2019) for extracting language features from radiology reports. 2. A text-guided low-resolution 3D diffusion model that jointly synthesizes CT volume and its anatomical structure volumes. 3. A super-resolution 3D diffusion model for scaling up the low-resolution generated volumes and complementing the missing anatomical details. First, we extract the shape information for the core anatomical structures, i.e., the lung lobes, the airway and the vessels. We choose the commonly used pre-trained segmentation tools to provide stable shape information for these three structures. Next, we propose an efficient hierarchical diffusion model with a two-phase process. In the first phase, we generate a low-

resolution volume ($64 \times 64 \times 64$) along with the anatomical structures conditioned via the radiology report. In the second phase, the model outputs a high-resolution volume $256 \times 256 \times 256$ along with the anatomical structures from a 3D super-solution diffusion model, which only takes low-resolution version as input. The low-resolution image ensures the volumetric consistency of the final images. We train our model on a large-scale 3D dataset, which contains 3D thorax computerized tomography (CT) images and associated radiology reports from 8,752 subjects. The dataset was collected by the University of Pittsburgh Medical Center and have been de-identified.

## Evaluation for Synthesis Quality

We compare our method with several baseline methods: current SOTA in GANs and Diffusion-based methods. Our method achieves better quality quantitatively overall.

Table 1: Quantitative comparison with different methods.

| Method | FID↓ | MMD↓ | Airway ($\times 10^4 mm^3$) |
|---|---|---|---|
| WGAN (Gulrajani et al. 2017) | 0.070 | 0.094 | $1.07_{\pm 0.64}$ |
| $\alpha$-GAN (Kwon, Han, and Kim 2019) | 0.028 | 0.057 | $1.14_{\pm 0.68}$ |
| HA-GAN (Sun et al. 2022) | 0.023 | 0.054 | $2.04_{\pm 0.73}$ |
| Medical Diffusion (Khader et al. 2023) | 0.013 | 0.022 | $1.77_{\pm 0.93}$ |
| **Ours** | **0.009** | **0.019** | $\mathbf{3.34_{\pm 1.19}}$ |
| **Ours** w/o shape | - | - | $1.99_{\pm 1.05}$ |
| Real | - | - | $4.58_{\pm 1.45}$ |

## Conclusion

Our research takes a significant step forward by synthesizing high-resolution 3D CT lung scans guided by detailed radiological and anatomical information. While previous models come with inherent limitations, particularly when generating intricate chest CT scan details. Our proposed MedSyn model addresses these challenges using a hierarchical training approach. Innovative architectural designs not only overcome previous constraints but also pioneer anatomy-conscious volumetric generation. Future work can leverage our model to enhance clinical applications.

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
