# OpenReview forum: "MedSyn: Text-guided Anatomy-aware Synthesis of High-Fidelity 3D CT Images"
_AAAI.org/2024/Spring_Symposium_Series/Clinical_FMs — AAAI 2024 SSS on Clinical FMs_

### Official Review · Reviewer_PLsW · 2024-02-21
**The paper introduces an innovative approach for generating high-quality 3D CT lung images using textual descriptions from radiology reports. This method leverages a hierarchical model that starts with low-resolution synthesis based on the textual input and progresses to higher resolutions, ensuring anatomical accuracy through segmentation masks. The model outperforms current generative models in terms of preserving crucial anatomical details and introduces a novel method for integrating textual information into the image generation process for medical imaging.**

**Rating:** 7
**Confidence:** 3

**Review:**

#### The approach involves a sophisticated hierarchical model that synthesizes high-resolution 3D medical volumes from textual descriptions found in radiology reports. This process is segmented into several core components, emphasizing the integration of a pre-trained text-encoder (Medical BERT) for linguistic feature extraction, a text-guided low-resolution 3D diffusion model for initial volume synthesis, and a super-resolution 3D diffusion model for enhancing detail and anatomical accuracy.


### Strength
#### 1. the paper utilizes a hierarchical diffusion model that begins with low-resolution image synthesis, advancing to detailed high-resolution outputs. This methodology ensures both the efficiency of computation and the integrity of generated volumetric data.

#### 2. By incorporating Medical BERT for text feature extraction, the model effectively leverages deep learning advancements in NLP to understand complex medical terminology and descriptions, enriching the synthesis process.

#### 3. The inclusion of segmentation masks for core anatomical structures (lung lobes, airway, vessels) within the synthesis process significantly enhances the anatomical plausibility and detail of the generated images, setting a new standard for medical image generation fidelity.

### Weakness

#### 1. The model's performance is directly tied to the quality and detail of the input radiology reports. In scenarios where such reports are lacking in detail or contain inaccuracies, the system's ability to generate precise and anatomically accurate images may be compromised.

#### 2. Despite efforts to manage memory use efficiently, the model's advanced capabilities and the need for high-resolution outputs inherently demand considerable computational resources, which may limit accessibility for some research and clinical settings.

### Questions:
#### 1. Could you elaborate on the steps you've taken to simplify the implementation of your hierarchical model for users without a deep technical background in AI or medical imaging?

#### 2. How does the model handle variations in the quality of textual descriptions in radiology reports? Are there mechanisms in place to ensure the reliability of image synthesis when faced with ambiguous or incomplete textual data?

#### 3. The paper focuses on lung CT images. Can the methodology be adapted for other types of medical imaging or conditions? If so, what modifications would be necessary?

#### 4. Could you discuss any measures taken to address potential biases in the datasets used for training the model? Additionally, how do you ensure the model's robustness across diverse patient demographics?

---

### Official Review · Reviewer_sSBD · 2024-02-21
**The paper presents a compelling approach using a two-step hierarchical diffusion model to generate high-fidelity 3D CT images while considering anatomical structures. Extending the results to showcase trade-offs with higher-quality volume generation could further enhance the paper's impact.**

**Rating:** 7
**Confidence:** 3

**Review:**

- The authors demonstrated how a two-step hierarchical diffusion model can generate anatomy-aware, high-fidelity 3D CT images.
- Generating anatomical structures along with the volume, conditioned on the radiology report, appears to be a promising approach.
- The paper is well-written and clearly explains the method.
- Figure 1 could be improved by providing explanations for each component.
- Extending the results to include higher-quality volume generation would help illustrate the trade-offs.

---

### Official Review · Reviewer_Ur87 · 2024-02-22
**Interesting method. Ablations on the impact of the different masks structures would have been interesting.**

**Rating:** 6
**Confidence:** 4

**Review:**

The paper presents a hierarchical volumetric text to image model. The authors integrate a use of mask annotations to stabilize the generation process to improve generated results. The model is trained on a custom dataset with pseudolabeled anatomical annotations. The metric scores on the custom dataset seem rather low thus some reference values to set the methods into perspective might be helpful. Additionally, it would be great to see the individual impact of each considered mask in the process. What happens when omitting what?
Overall the idea seems nice and interesting and as such i tend towards accepting the paper.

---

### Official Review · Reviewer_Pe4N · 2024-02-23
**Good approach to CT generation, more motivation behind design decisions needed**

**Rating:** 6
**Confidence:** 3

**Review:**

This paper introduces MedSyn, a new hierarchical diffusion model approach for generating CTs. While this is a valuable contribution, more motivation behind design decisions are needed. For example, why isn't latent diffusion needed? What differs from medical diffusion and why does that lead to better performance? Evaluation on more standard public datasets would also be ideal for this paper for more systematic comparison (Medical Diffusion uses LIDC-IDRI for example).